# TIPS: Two-Level Prompt for Rehearsal-free Continual Learning

## Abstract

Continual learning based on prompt tuning creates a key-value pool, where these key-value pairs are called prompts. Prompts are retrieved using input images as queries and input into a frozen backbone network. It requires training only a few parameters to quickly adapt to downstream tasks. Compared to other traditional continual learning methods, it is more effective in resisting catastrophic forgetting. However, the effectiveness of these methods heavily depends on the selection strategy. Most existing methods overlook the model plasticity since they focus on solving the model's stability issues, leading to a sharp decline in performance for new tasks in long task sequences of incremental learning. To address these limitations, we propose a novel prompt-based continual learning method called TIPS, which mainly consists of two modules: (1) design a novel two-level prompt selection strategy combined with a set of adaptive weights for sparse joint tuning, aiming to improve the accuracy of prompt selection; (2) design a semantic distillation module that enhances the generalization ability to unknown new classes by creating a language token and utilizing the encapsulated semantic information of class names. We validated TIPS on four datasets across three incremental scenarios. Our method outperformed the current state of the art (SOTA) by 2.03%, 4.78%, 1.18%, and 5.59% on CIFAR (10 tasks), ImageNet-R (20 tasks), CUB (10 tasks), and DomainNet (20 tasks). Notably, our approach consistently surpasses or matches SOTA in all settings, maintaining stable prompt selection accuracy throughout multiple incremental learning sessions.

## 1 Introduction

Humans are good at gaining new knowledge while remembering past information. However, machine learning finds it hard to copy this ability (Van de Ven & Tolias, 2019; Masana et al., 2022). Neural networks often forget learned knowledge when learning new tasks, which is called catastrophic forgetting (De Lange et al., 2021; McCloskey & Cohen, 1989). Continual learning (CL) (Mehta et al., 2023; Masana et al., 2022) achieves great success in solving catastrophic forgetting and has received increasing attention in recent years (Parisi et al., 2019). To handle catastrophic forgetting, CL aims at smoothly integrating new tasks into a single model while preventing catastrophic forgetting of previously learned knowledge. However, maintaining this balance (*i.e.,* the stability-plasticity dilemma (Arani et al., 2022; Wang et al., 2022a)) poses great challenges for effective CL.

Traditional from-scratch training methods in CL aim to prevent knowledge forgetting by protecting important past parameters. For example, Elastic Weight Consolidation (EWC) (Kirkpatrick et al., 2017), uses regularization constraints on the loss function for new tasks to protect previously acquired knowledge from being disrupted by new information. DER (Yan et al., 2021) employs a dynamic network architecture that creates a distinct parameter space for each task, while freezing the existing parameters to ensure they remain unchanged, thereby achieving effective parameter isolation. However, these methods not only require multiple iterations to converge but also rely on rehearsal strategy[1] to perform well. It poses significant challenges in scenarios that require rapid generalization or involve privacy concerns.

---

[1] Allocate extra storage space for several old task examplars. When learning a new task, learn all the examplars from the storage space together with the new task examples.

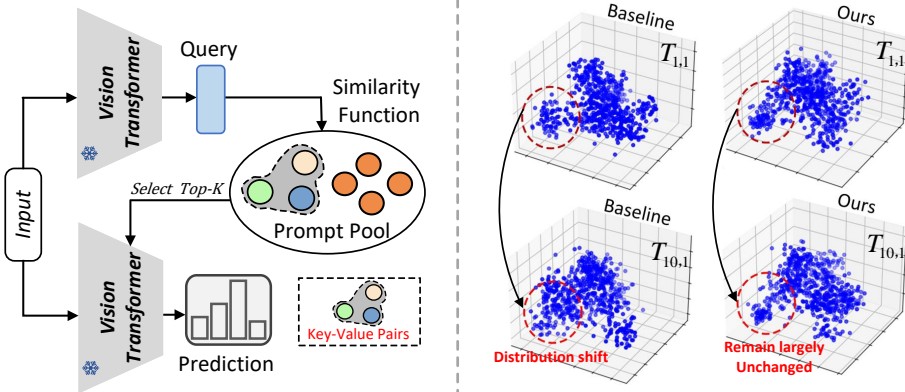

Figure 1: **Left**: The framework of previous prompt-based CL. They typically use a Vision Transformer (ViT) (Dosovitskiy et al., 2020) as the feature extractor, treating the input image features as the query. A similarity function (*e.g.,* cosine similarity) is then used to directly select the most similar prompts (key-value pairs) from the prompt pool based on the query. Finally, the selected prompt is input into the backbone network along with the image. However, the accuracy of selecting the correct prompt using these simple selection strategies declines sharply, leading to a decrease in model stability in long-sequence task settings. **Right**: 3D-KDE analysis of ImageNet-R. $T_{1,1}$ represents the density distribution of all samples in Task- 1 after completing the first task, while $T_{10,1}$ represents the density distribution of all samples in Task-1 after completing the 10-th task. At the completion of the first task, the model has the optimal understanding of Task- 1. If the density distribution changes less after $n$ rounds of incremental learning sessions, the model is considered more stable. We present the density distributions of the baseline and our method, showing that the density distribution changes less with our method, indicating a more stable model.

A recent innovation in CL influenced by prompt learning is referred to as prompt-based pre-trained CL approaches. In training phase, these methods (Wang et al., 2022b;a; Chen et al., 2023) first freeze the pre-trained Transformer backbone and then train the corresponding prompts to adapt to specific downstream tasks. Each task generally has a unique set of prompts, and all of the prompt sets constitute a prompt pool. A similarity function is employed to select the prompt that best matches the current test example during inference phase. The selected prompt is then concatenated with the example's embedding and input into the backbone network. These methods significantly reduce the impact of catastrophic forgetting due to extensive prior knowledge of the pre-trained model and allow the model to converge with very few iterations.

Although important progress has been achieved by existing prompt-based CL methods, they have the following drawbacks to address: (1) Most of prompt-based CL methods (*e.g.,* (Wang et al., 2022b; Chen et al., 2023)) heavily depend on selection strategy. During the inference phase, task identifiers are not available, so only the selection strategy can identify the appropriate prompt from training. Conventional selection methods (as shown in Figure 1) that measure similarities between images and prompts show a significant decrease in the probability of accurately selecting the correct prompt as the number of task sessions and prompts increases. (2) Many prompt-based CL methods (*e.g.,* (Wang et al., 2023a; Smith et al., 2023)) focus on solving the forgetting problem (stability) while neglecting the model's generalisation ability (plasticity), resulting in a limited understanding of new tasks by specific task prompts within a few model iterations.

In this paper, we propose a novel prompt-based CL framework, as illustrated in Figure 2. Our goals include: (1) designing a stable prompt selection strategy to prevent a significant decline in accuracy when selecting the correct prompt as the size of the prompt set increases; and (2) developing a simple and effective module to enhance the model's understanding of new tasks and improve its plasticity. To achieve these goals, we construct a two-level prompt. The first-level prompt consists of a set of learnable parameters and text embeddings of class labels, it improves the probability of matching image embeddings with class prototypes by learning the high-level semantic features of images, thereby addressing the instability of the prompt selection strategy. Moreover, we propose a set of adaptive weight parameters, with each weight corresponding to a prompt, which dynamically

adjusts the importance of each prompt across different tasks. These weights capture the underlying complexity of various tasks during forward propagation, enabling the linear combination and passing of the prompt set to any depth of the backbone network, thereby enhancing the model's resistance to forgetting old classes. Finally, we introduce a semantic knowledge distillation module that incorporates a learnable semantic token to capture the semantic information of current class labels, aiding second-level prompts in understanding new tasks and improving the model's plasticity.

Compared to previous work, contributions of our work include:

- We designed a two-level prompt selection strategy that leverages the semantic consistency between images and class labels to improve the accuracy of selecting the corresponding prompt. This strategy further adjusts the relevant prompts using adaptive weighting. Compared to traditional selection methods, our approach reduces potential errors in prompt selection and enhances stability during long incremental task sessions.

- We designed a semantic distillation module that integrates visual and linguistic modalities to utilize the semantics of the original class labels in text form. This simple and novel method enhances plasticity by focusing on learning the knowledge of the current task without relying on intricately designed boosters, effectively alleviating the issue of insufficient understanding of new tasks.

- Our method is not restricted by specific datasets or incremental scenarios due to the well-designed framework. We conduct ablation studies in detail, including the loss function and proposed structures to understand the model. Extensive experiments demonstrate that it achieves state-of-the-art CL performance.

## 2 RELATED WORK

**Continual learning from scratch.** The first category is regularization-based (Wu et al., 2019; Kirkpatrick et al., 2017; Chaudhry et al., 2018) CL, where the primary idea is to employ a penalty mechanism to ensure that important parameters remain unchanged. The second category is replay-based (Rebuffi et al., 2017; Li & Hoiem, 2017; Feng et al., 2024a; Douillard et al., 2020) CL, which allocates extra storage space for a few exemplars of previous tasks. When learning a new task, the system learns all the exemplars from the storage space, as well as the examples from the new task. The third category is parameter-isolation-based (Feng et al., 2024b; Yan et al., 2021; Douillard et al., 2022) CL, which allocates independent learning parameters for each task. While some methods employ sophisticated compression strategies to reduce the number of model parameters, they often neglect rapid generalization. Consequently, these models usually require a large number of iterations to converge.

**Prompt-Based Continual Learning.** Prompt-based CL keeps the pre-trained model's weights unchanged while adding additional learnable prompt tuning modules to generalize to downstream tasks. These works (Wang et al., 2022b;a; Smith et al., 2023) create a prompt pool where each prompt contains a learnable index key and a prompt value. They uses a cosine distance function to search for the nearest in the existing pool, then optimizes weights using cross-entropy loss. In recent, some works (Zhou et al., 2022; Wang et al., 2023a) integrated CLIP (Radford et al., 2021) with a backbone network to form a framework, inspiring recent innovations. For example, Chen et al. (2023) enhances model diversity using CLIP to replace identical pre-trained models for constructing different classifiers, dynamically combining logits during inference for comprehensive decision-making. Zheng et al. (2023) applied merging techniques to CLIP models to maintain their zero-shot performance during CL. Although this merging method does not require saving all historical models, deciding which parameters to merge remains a challenge.

The key difference between our method and the aforementioned methods is that our approach does not require selecting or merging parameters. Our method and others are based on the same pretrained backbone (*e.g.,* CLIP encoder). However, our approach achieves comparable or even better results without necessitating fine-tuning the parameters of the backbone network. Moreover, our work does not require any additional storage space for previous instances of old classes by leveraging the prior knowledge of the powerful pre-trained model, making our approach suitable for data privacy scenarios.

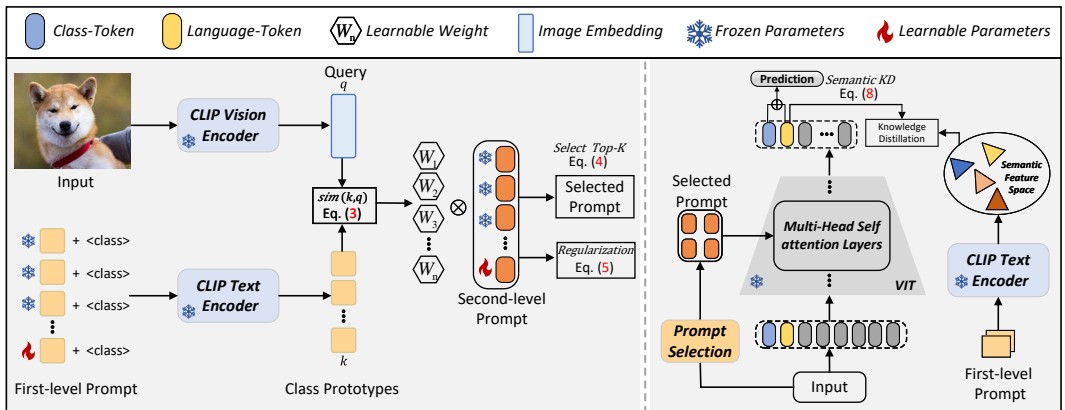

Figure 2: **Left**: Our proposed prompt selection strategy uses a two-level prompt architecture. The first-level generates class prototypes as keys, which are compared with the query for similarity. Then, the optimal prompt is selected by incorporating learnable parameters. Notably, we freeze the prompt parameters from previous tasks when we fine-tune the prompts for the current task to prevent them from being affected by the current task. **Right**: Overall framework of our method. Selected prompts and image embeddings are fed into the pre-trained Vision Transformer (VIT) model, which employs prefix tuning to adapt to downstream tasks. To tackle the plasticity challenge inherent in prompt-based CL, the CLIP text encoder is harnessed to extract the semantics of class names and images. Subsequently, semantic knowledge distillation is applied to facilitate back-propagation into the model and language tokens.

## 3 METHODOLOGY

### 3.1 PRELIMINARY

**Problem Definition.** The goal of CL is to acquire knowledge from a stream of data composed of $T$ non-overlapping sequential datasets, denoted as $\mathcal{D} = \{\mathcal{D}_1, \mathcal{D}_2, \ldots, \mathcal{D}_t\}$. Each dataset $\mathcal{D}_t$ corresponds to a specific task $t$, which can be represented as a collection of data for that specific class, *i.e.*, $\mathcal{D}_t = \bigcup_i (x_c^t, y_c^t)$, contains data samples $x_c^t \in X$ and corresponding $y_c^t \in Y$, where $c$ denotes the $c$-th class within task $t$, The objective is to train a mapping function $f_\theta : X \to Y$ parameterized by $\theta$ to handle the $T$ incremental tasks. In the inference phase, $f$ predicts the corresponding label $y$ based on task-agnostic samples $x$. Note that during the training phase of the current task, data from previous tasks is not accessible.

### 3.2 OVERALL FRAMEWORK

In our work, we introduce a new prompt-based CL method called TIPS, which aims to address the instability of the prompt selection strategy and the low plasticity of the model. Specifically, TIPS consists of two modules: Two-level prompt (TP) and semantic knowledge distillation. As shown in Figure 2, TP is based on a two-level prompt selection strategy. The first-level prompt generates a key for retrieval by combining class labels with learnable context parameters and feeding them into the CLIP text encoder. Then, the visual embedding of the input image are used as a query, which is matched with the key. Finally, we apply adaptive weight modulation to the second-level prompt, which enables joint sparse prompt tuning, thereby reducing prompt selection errors in long incremental task sequences. To facilitate the model's adaptation to CL incremental tasks, a semantic knowledge distillation module was proposed. The semantic knowledge of the current class is transferred to a learnable language token by using a simple distillation function.

### 3.3 TWO-LEVEL PROMPT

To improve pertained model's performance on downstream tasks, previous methods (*e.g.,* Coop (Zhou et al., 2022)) use a set of learnable vectors to replace the preset templates (*e.g.,* a photo of [cat]). The key point of those method is encoding the knowledge of the downstream data into these

vectors to guide the model in adapting to downstream tasks. Specifically, the learnable contextual vectors $\mathbf{p}$ is concatenated with class name $y_c \in Y_t$, where the text description of the $c$-th class is:

$$\mathbf{P}_c = ([\mathbf{p}]_1; [\mathbf{p}]_2; \dots [\mathbf{p}]_m; [\mathbf{CLS}]_c), \tag{1}$$

where each $[\mathbf{p}]_i \in \mathbb{R}^D$, $i \in \{1, 2 \dots, m\}$, $[\mathbf{CLS}]_c$ is the text embedding of the $c$-th class name. COOP uses the embedding output $\mathbf{k}$ as a class prototype for classification, feed an image $\mathbf{x} \in \mathbb{R}^{H \times W \times C}$ and a text embedding $\mathbf{P}_c$ into the CLIP image encoder $E_{vis}$ and text encoder $E_{txt}$, yielding the image embedding $\mathbf{q} = E_{vis}(\mathbf{x})$ and the class prototype $\mathbf{k}_c = \frac{1}{N} \sum_{i=1}^{N} E_{txt}(\mathbf{P}_c)$. Consequently, it can calculate the probability of class $y_c$ prediction for the test image $\mathbf{x}$ using the following formula:

$$p(y_c \mid \mathbf{x}) = \frac{e^{\langle \mathbf{q}, \mathbf{k}_c \rangle / \tau}}{\sum_{i=1}^{N} e^{\langle \mathbf{q}, \mathbf{k}_c \rangle / \tau}}, \tag{2}$$

where $\tau$ is a temperature parameter learned by CLIP, $\langle \cdot, \cdot \rangle$ denotes the cosine distance similarity, $\mathbf{k}_c$ is the embedding of $c$-th class in current task, and $N$ is the total number of classes. In our method, we use $\mathbf{P}$ as the first-level prompt and the output $\mathbf{k}$ from $E_{txt}$ as the class prototype key in order to enhance the accuracy of selecting corresponding prompts. In particular, we calculate similarity scores by matching the queries $\mathbf{q}_c^t$ with the keys $\mathbf{k}_c^t$ to retrieve the second-level prompt:

$$\mathbf{S}_c^t = \langle \mathbf{q}_c^t, \mathbf{k}_c^t \rangle, \tag{3}$$

where $c$ represents the class encoding in the current task, and $t$ represents the task encoding. The similarity scores are then projected onto shared space $\mathbf{S}_t \in \mathbb{R}^{N \times D}$. To make prompt selection more flexible, we propose a set of adaptive weights $\mathbf{W} \in \mathbb{R}^{N \times D}$ to regulate the correlation between the task and the prompts, achieving joint sparse prompt tuning. The final output of the second-level prompt as follow:

$$\hat{\mathbf{P}}_c^t = \text{TOP-K}^{max}\{\mathbf{S}_c^t \cdot \mathbf{W} \cdot \hat{\mathbf{P}}_c^t\}. \tag{4}$$

Each class corresponding to each prompt, *i.e.,* $\hat{\mathbf{P}}_t = [\hat{\mathbf{P}}_1; \hat{\mathbf{P}}_2; \dots \hat{\mathbf{P}}_N]$, where $\hat{\mathbf{P}} \in \mathbb{R}^{N \times M \times D}$, $M$ denotes length of second prompt. In addition, we also follow CODA-Prompt (Smith et al., 2023) regularisation penalty to prevent potential similarities between the new and old prompts, it provides more diversity and less homogeneity in prompts:

$$\mathcal{L}_O = \sum_{t=T_i, t' \in \text{past}(T_i)} \left\| \hat{\mathbf{P}}_t, \hat{\mathbf{P}}_{t'} \right\|_2, \tag{5}$$

where $T_i$ denotes as current task, and $\text{past}(t) = \{t' | t' \in T, t' < i\}$ represents previous learned task.

From a high-level perspective, the features extracted by $E_{vis}$ contain rich advanced information, such as image labels and contextual details. By optimizing the first-level prompt, the text's contextual parameters can better learn the high-level semantics of the image. As a result, this strategy makes the query and key features more similar, increasing the success rate of matching. Furthermore, to retain this capability, we freeze the learnable parameters from previous tasks while learning the current task, ensuring more stable prompt selection in long task sequence settings.

## 3.4 SEMANTIC KNOWLEDGE DISTILLATION

The implementation of a two-level prompt with adaptive weight for prompt selection enables our model to effectively adapt to downstream tasks, ensuring high stability. However, a highly stable model has an impact on new classes' learning. Therefore, we propose a semantic knowledge distillation module to capture the semantic usage of new class names, enhancing the model's ability to learn new classes.

To capture the textual semantic information of class names, we created a language token $\ell_i \in [\ell_1, \ell_2, \dots, \ell_D]$, as shown in Figure 2. The language token embedding $\mathbf{e}_c^\ell = f_\theta(\ell_c)$ and class token $\zeta_i \in [\zeta_1, \zeta_2, \dots, \zeta_D]$ embedding $\mathbf{e}_c^\zeta = f_\theta(\zeta_c)$ is summed to form a new embedding for $c$-th class:

$$\mathbf{e}_c^{sum} = \alpha \cdot \mathbf{e}_c^\ell + \beta \cdot \mathbf{e}_c^\zeta, \tag{6}$$

where $\alpha$ and $\beta$ are hyperparameters that regulate the integrate process, we used $\alpha = 0.5$ and $\beta = 0.5$ for all of our experiments, the final embedding $e_c^{sum}$ is passed through softmax to generate the final prediction:

$$\hat{y}_c = \text{Softmax}(f_\theta(\mathbf{e}_c^{sum})). \tag{7}$$

Eq.6 and Eq.7 can effectively enhance the understanding of the current task by integrating the two modalities' information at the decision layer. However, their output features become similar because the two tokens share the same goal in the training phase. It hinders the effective training of the language token. To enhance the discriminative ability of the language token, we propose a new loss function called semantic distillation ($\mathcal{L}_{\mathcal{SD}}$).

The text encoder of CLIP is utilized to extract the embedding feature of the class-name and contextual information of image $\mathbf{h}_c = E_{txt}\left(\mathbf{P}_c\right), \mathbf{h}_c \in \mathbb{R}^D$. Simultaneously, we use the ViT backbone to obtain the output feature $\mathbf{e}_c^\ell$ corresponding to the language token. Thereby, we use a simple distillation function (Hinton et al., 2015) to extract high-level information of image from semantic feature space into the language token feature:

$$\mathcal{L}_{\mathcal{SD}} = \mathcal{L}_{\text{CrossEntropy}}\left(\hat{y}_c, y_c\right) + \lambda \cdot \mathcal{L}_{KD}\left(\mathbf{e}_c^\ell, \mathbf{h}_c\right), \tag{8}$$

where $\lambda$ is the balancing hyperparameter, $\hat{y}_c$ denotes the prediction result of the classification head. For detailed experimental settings, please refer to Table 10.

Table 1: The average accuracy (%) and the number of fine-tuned parameters for 4 datasets with an incremental number of 10 tasks are presented, with DomainNet being a cross-domain dataset. Methods are grouped according to the buffer size, where 0 indicates no replay is needed, * denotes results directly taken from the original paper, and - indicates experiments that could not be completed, the rest of the experiments were conducted using the code provided by their original paper. We conducted experiments using three random seeds: 1993, 1997 and 1999.

| Model | Buffer size | CIFAR-100 | ImageNet-R | CUB-200 | DomainNet | #Para(M) |
|---|---|---|---|---|---|---|
| Joint-Training (*ViT-B/16*) | 0 | 93.23 | 90.38 | 88.13 | 89.15 | 86.00 |
| Fine-Tune (*ViT-B/16*) | 0 | $18.42_{\pm 0.23}$ | $18.87_{\pm 2.65}$ | $18.52_{\pm 1.99}$ | $10.68_{\pm 3.25}$ | 86.00 |
| Experience Replay (*ViT-B/16*) | 5000 | $82.53_{\pm 0.17}$ | $65.18_{\pm 0.40}$ | $63.12_{\pm 1.44}$ | $59.23_{\pm 1.32}$ | 86.00 |
| EWC (Kirkpatrick et al., 2017) | 2000 | $54.14_{\pm 1.25}$ | $40.27_{\pm 3.24}$ | $38.25_{\pm 2.45}$ | - | 86.00 |
| LwF (Li & Hoiem, 2017) | 2000 | $20.35_{\pm 0.98}$ | $20.48_{\pm 0.58}$ | $17.45_{\pm 1.27}$ | - | 86.00 |
| Gdumb (Prabhu et al., 2020) | 2000 | $67.14_{\pm 0.37}$ | $44.28_{\pm 0.51}$ | $61.34_{\pm 0.46}$ | - | 86.00 |
| DER++ (Yan et al., 2021) | 2000 | $61.06_{\pm 0.75}$ | $57.64_{\pm 0.98}$ | $75.84_{\pm 1.35}$ | - | 86.00 |
| PromptFusion (Chen et al., 2023) | 2000 | $87.40_{\pm 0.00}$[*] | $80.70_{\pm 0.00}$[*] | - | - | 0.35 |
| PC (Dai et al., 2024) | 0 | $88.04_{\pm 0.00}$[*] | $74.34_{\pm 0.00}$[*] | - | - | - |
| L2P (Wang et al., 2022b) | 0 | $89.24_{\pm 0.04}$ | $76.82_{\pm 0.38}$ | $77.28_{\pm 0.72}$ | $71.63_{\pm 0.95}$ | 0.20 |
| Dualprompt (Wang et al., 2022a) | 0 | $87.39_{\pm 0.35}$ | $73.97_{\pm 0.44}$ | $79.14_{\pm 0.58}$ | $71.98_{\pm 0.99}$ | 0.41 |
| AttriCLIP (*ViT-L/14*) (Wang et al., 2023a) | 0 | $81.40_{\pm 0.00}$[*] | $81.71_{\pm 0.35}$ | $58.53_{\pm 1.47}$ | $74.59_{\pm 1.02}$ | - |
| CODA-Prompt (Smith et al., 2023) | 0 | $90.40_{\pm 0.12}$ | $78.69_{\pm 0.48}$ | $81.05_{\pm 0.33}$ | $81.41_{\pm 0.82}$ | 3.91 |
| ESN (Wang et al., 2023b) | 0 | $90.38_{\pm 0.67}$ | $73.66_{\pm 0.92}$ | $83.01_{\pm 1.00}$ | $82.99_{\pm 1.18}$ | 3.07 |
| Ours | 0 | $\mathbf{92.43}_{\pm 0.34}$ | $\mathbf{83.77}_{\pm 0.67}$ | $\mathbf{84.29}_{\pm 0.69}$ | $\mathbf{88.39}_{\pm 0.78}$ | 3.61 |

Table 2: Result of the forgetting rate (%) ↓ (Chaudhry et al., 2018) for 4 datasets with an incremental number of 10 tasks, where lower values are better.

| Model | CIFAR-100 | ImageNet-R | CUB-200 | DomainNet |
|---|---|---|---|---|
| Fine-Tune (*ViT-B/16*) | 88.68 | 84.72 | 90.64 | 94.35 |
| Experience Replay | $16.46_{\pm 0.25}$ | $23.31_{\pm 0.89}$ | $24.32_{\pm 0.22}$ | $27.18_{\pm 1.23}$ |
| LwF (Li & Hoiem, 2017) | $87.23_{\pm 2.34}$ | $80.12_{\pm 1.02}$ | $83.81_{\pm 0.95}$ | - |
| DER++ (Yan et al., 2021) | $39.87_{\pm 0.99}$ | $39.12_{\pm 1.01}$ | $28.18_{\pm 0.79}$ | - |
| PC (Dai et al., 2024) | $5.61_{\pm 0.00}$[*] | $7.35_{\pm 0.00}$[*] | - | - |
| L2P (Wang et al., 2022b) | $7.63_{\pm 0.30}$[*] | $4.22_{\pm 0.47}$ | $13.60_{\pm 0.28}$ | $15.98_{\pm 1.25}$ |
| Dualprompt (Wang et al., 2022a) | $6.71_{\pm 0.09}$[*] | $4.68_{\pm 0.20}$[*] | $11.04_{\pm 0.26}$ | $9.16_{\pm 0.31}$ |
| AttriCLIP (*ViT-L/14*) (Wang et al., 2023a) | $10.32_{\pm 0.21}$ | $6.44_{\pm 0.11}$ | $30.17_{\pm 0.14}$ | $22.35_{\pm 3.34}$ |
| CODA-Prompt (Smith et al., 2023) | $7.03_{\pm 0.34}$ | $4.88_{\pm 0.30}$ | $6.91_{\pm 0.12}$ | $10.15_{\pm 0.12}$ |
| ESN (Wang et al., 2023b) | $5.44_{\pm 0.48}$ | $5.20_{\pm 0.77}$ | $6.73_{\pm 1.77}$ | $10.62_{\pm 2.12}$ |
| Ours | $\mathbf{5.05}_{\pm 0.70}$ | $\mathbf{3.63}_{\pm 0.24}$ | $\mathbf{6.00}_{\pm 1.32}$ | $\mathbf{8.89}_{\pm 0.86}$ |

## 4 EXPERIMENTS

### 4.1 EXPERIMENTAL SETTINGS

**Datasets.** A large number of experiments were conducted on 4 benchmark datasets to thoroughly compare different continual learning methods. Note that all test samples do not have task identifiers. We follow (Smith et al., 2023; Wang et al., 2022b; 2023b) to evaluate CIFAR-100 (Krizhevsky et al., 2009), ImageNet-R (Hendrycks et al., 2021), and DomainNet (Peng et al., 2019). Furthermore, we included the CUB-200 (Wah et al., 2011) dataset, which has markedly distinct classes from the backbone network's pre-training dataset, thereby providing an efficient evaluation of the model's adaptability to downstream tasks.

**Protocols.** We divided each dataset into 5, 10, and 20 incremental task sessions, which means that the number of classes per incremental session is the total number of classes in each dataset divided by 5, 10, and 20, respectively. It is worth noting that DomainNet collected 345 different classes from six domains. We followed the ESN (Wang et al., 2023b) division rule and selected 200 classes to form a new dataset.

**Implementation Details.** To ensure a fair comparison, all methods use the ImageNet-21K pretrained VIT-B-16 (Dosovitskiy et al., 2020) as the backbone network. We optimise our model with a learning rate of 0.001 and set the number of epochs to 20 using Adam (Kingma & Ba, 2014). We set the pool size to 200 for all datasets, with the exception of CIFAR, which we set to 100. We set the second-level prompt $M$ length to 4 and the first-level prompt $m$ length for CLIP to 16. In the first level of prompts, we initialize the prefix parameters with tokenize of "XXXX". In the second-level prompts, we randomly initialize the parameters. For the CLIP setup, we follow AttriCLIP (Wang et al., 2023a) setting and choose ViT-L-14 as the backbone network. For a detailed explanation, please refer to Algorithm 1.

**Metrics.** We denote the TOP-1 accuracy after task $t$ as $A_t$, and $Last$ represents the accuracy of the last task. The average accuracy is denoted as $\text{Avg} = \sum_{t=1}^{n} A_t$.

Table 3: Result of average accuracy Avg and accuracy of last task on 5 tasks setting. All experiments were conducted in 1993 random seed.

| Method | CIFAR | | ImageNet-R | | CUB | | DomainNet | |
|---|---|---|---|---|---|---|---|---|
| | Avg | Last | Avg | Last | Avg | Last | Avg | Last |
| L2P (Wang et al., 2022b) | 90.61 | 85.98 | 77.69 | 73.95 | 79.96 | 71.46 | 76.94 | 71.10 |
| Dualprompt (Wang et al., 2022a) | 89.99 | 85.63 | 75.21 | 70.88 | 80.49 | 72.09 | 73.69 | 69.52 |
| AttriCLIP (*ViT-L/14*) (Wang et al., 2023a) | 82.68 | 76.12 | 81.78 | 78.23 | 65.53 | 55.37 | 73.36 | 66.11 |
| CODA-Prompt (Smith et al., 2023) | 92.20 | 88.64 | 79.23 | 74.88 | 83.40 | 81.17 | 86.60 | 79.23 |
| ESN (Wang et al., 2023b) | 91.71 | 88.51 | 75.30 | 71.02 | **85.91** | **84.82** | 84.31 | 75.91 |
| Ours | **92.43** | **89.04** | **85.11** | **82.97** | 85.03 | 83.93 | **88.78** | **83.11** |

Table 4: Result of average accuracy Avg and accuracy of last task on 20 tasks setting. All experiments were conducted in 1993 random seed.

| Method | CIFAR | | ImageNet-R | | CUB | | DomainNet | |
|---|---|---|---|---|---|---|---|---|
| | Avg | Last | Avg | Last | Avg | Last | Avg | Last |
| L2P (Wang et al., 2022b) | 84.18 | 77.72 | 74.50 | 69.87 | 70.07 | 57.80 | 66.46 | 58.54 |
| Dualprompt (Wang et al., 2022a) | 83.65 | 77.91 | 71.22 | 65.15 | **78.98** | 66.16 | 63.21 | 55.36 |
| AttriCLIP (*ViT-L/14*) (Wang et al., 2023a) | 79.54 | 62.08 | 77.12 | 71.75 | 60.60 | 43.25 | 72.33 | 61.94 |
| CODA-Prompt (Smith et al., 2023) | **88.52** | **83.42** | 74.80 | 69.80 | 72.11 | 63.40 | 80.78 | 67.91 |
| ESN (Wang et al., 2023b) | 87.15 | 80.37 | 70.46 | 64.28 | 65.69 | 63.10 | 79.59 | 66.19 |
| Ours | 87.46 | 81.90 | **80.39** | **77.90** | 75.75 | **70.78** | **85.56** | **74.42** |

### 4.2 EXPERIMENTAL RESULTS.

We compared classical methods with recent state-of-the-art (SOTA) methods: EWC (Kirkpatrick et al., 2017), LWF (Li & Hoiem, 2017), Gdumb (Prabhu et al., 2020), Der++ (Yan et al., 2021), Promptfusion (Chen et al., 2023), L2P (Wang et al., 2022b), Dualprompt (Wang et al., 2022a),

AttriClip (Wang et al., 2023a), Coda-Prompt (Smith et al., 2023), ESN (Wang et al., 2023b) and PC (Dai et al., 2024). We also reported the baseline method, which involves sequential fine-tuning and joint-training of a pre-trained model.

**Main Result.** We present the results of various methods on four datasets—CIFAR, CUB, ImageNet-R, and Domainnet for 5, 10, and 20 tasks in Tables 1, 3, and 4, respectively. Overall, our method achieves excellent performance compared to the recent SOTA methods across the four datasets without replaying old classes examplars. Additionally, our method also shows a competitive number of tuned parameters. One could find that:

(1) TIPS demonstrated strong performance across various metrics. Specifically, in the 10 tasks setting, TIPS outperformed the best comparative method, ESN (which achieved 83% on DomainNet), by approximately 5% in terms of the Avg metric. This improvement is attributable to our prompt selection strategy, which relies on the similarity between image queries and class prototype keys both share similar high-level semantic information, thereby enhancing the stability of the selection process.

(2) Our methods have advanced the addressing of the stability-plasticity dilemma beyond the achievements of previous prompt-based CL methods. Although TIPS is also a prompt-based CL approach, it achieves substantially better performance on the LAST metric, outperforming recent SOTA methods by approximately 4–6%. This improvement is attributed to our introduction of language tokens, which enable the framework to more effectively learn the semantic information of newly introduced classes.

(3) Our method demonstrates significant advancements on more challenging cross-domain datasets like ImageNet-R and DomainNet, despite only achieving modest improvements on simpler tasks. Specifically, TIPS achieves a performance improvement over the second-best method by 5.59% on ImageNet-R (20 tasks) and by 4.78% on DomainNet (20 tasks) on long sequences setting.

To further verify our method, the forgetting rates for the 10-task scenario are reported in Table 2. Although a rehearsal strategy is not employed, a lower forgetting rate across all four datasets compared to recent SOTA methods is exhibited by our approach. Additionally, experiments were conducted on the CUB dataset, which has a distribution vastly different from that of the pre-trained backbone. In the 10-task scenario, an improvement over ESN by 1.18% is achieved by our method. This suggests that performance of our method is independent of any particular dataset or CL environment.

Table 5: Result of ablation study on 10 tasks setting. Last-T denotes the TOP-1 accuracy for the new classes of the last task.

| TP | AW | $\mathcal{L}_{\mathcal{SD}}$ | Cifar-100 | | ImageNet-R | | CUB-200 | | DomainNet | |
|----|----|------|-----------|--------|------------|--------|---------|--------|-----------|--------|
|    |    |      | Avg | Last-T | Avg | Last-T | Avg | Last-T | Avg | Last-T |
|    |    |      | 86.99 | 79.35 | 75.36 | 72.36 | 75.38 | 84.35 | 77.36 | 31.35 |
| ✓  |    |      | 90.94 | 85.00 | 80.36 | 75.78 | 81.40 | 87.35 | 83.69 | 42.31 |
|    | ✓  |      | 90.85 | 86.10 | 78.93 | 75.78 | 80.65 | 91.02 | 82.64 | 30.33 |
| ✓  | ✓  |      | 91.64 | 88.20 | 83.03 | 82.57 | 82.90 | 88.57 | 87.42 | 46.11 |
| ✓  | ✓  | ✓    | **92.43** | **91.10** | **83.77** | **85.67** | **84.29** | **94.69** | **88.39** | **57.94** |

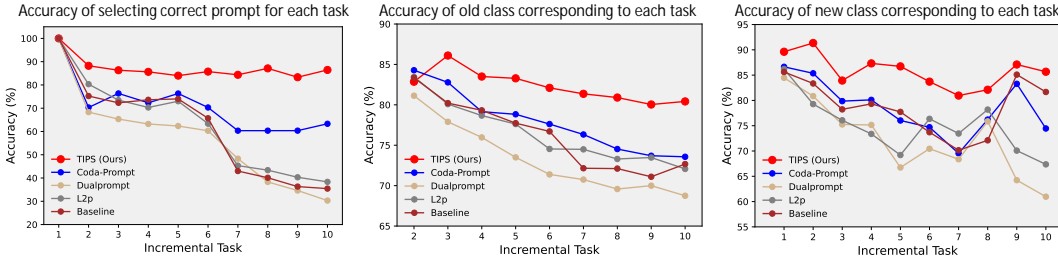

Figure 3: **Left**: Accuracy of selecting correct prompts on training for each task across various incremental tasks with different methods. **Middle**: The learning accuracy of old classes in each task varies with different methods. **Right**: The learning accuracy of new classes in each task varies with different methods. Note: All experiments were conducted on ImageNet-R 10 tasks.

Table 6: Result of applying TP and the $\mathcal{L}_{\mathcal{SD}}$ to L2p. The experiment was conducted on ImageNet-R 10 tasks setting.

|  | Avg | Last-T |
|---|---|---|
| L2p | 77.20 | 71.62 |
| L2p+TP | 80.28 | 73.32 |
| L2p+$\mathcal{L}_{\mathcal{SD}}$ | 79.26 | 77.32 |

Table 7: Result of applying TP and the $\mathcal{L}_{\mathcal{SD}}$ to CODA-Prompt. The experiment was conducted on ImageNet-R 10 tasks setting.

|  | Avg | Last-T |
|---|---|---|
| CODA-Prompt | 79.17 | 74.45 |
| CODA-Prompt+TP | 81.70 | 75.32 |
| CODA-Prompt+$\mathcal{L}_{\mathcal{SD}}$ | 81.06 | 81.39 |

### 4.3 Ablation Study

(1) **Structural analysis.** The proposed method TIPS consists of two main modules: two-level prompt (TP) selection strategy combined with a set of adaptive weights (AW); and semantic knowledge distillation module ($\mathcal{L}_{\mathcal{SD}}$). The impact of each part on the four datasets is shown in Table 5. First, the results in the first three rows indicate that both TP and AW improve the Avg and Last-T metrics compared to the baseline. Second, the combination of TP and AW results in the highest increase in the term of Avg in the fourth row. For instance, across all datasets, it achieves an improvement of 4.65%, 7.67%, 7.52%, and 10.06% compared to baseline. Finally, it can be observed that the sustained performance increase is manifested in terms of Avg and Last-T with the incorporation of the $\mathcal{L}_{\mathcal{SD}}$ module (as shown in fifth row). For example, it achieves an improvement of 26.59% in term of Last-T compare to the baseline in the DomainNet dataset. Overall, our method, which incorporates all three modules (as shown in the last row), achieved the best results. Therefore, the experiments validate the effectiveness of the proposed three modules.

(2) **Selection strategy analysis.** We report the performance of the prompt selection strategy and the results on both old and new classes for TIPS compared to other methods in Figure 3. In the left figure, the performance of our method can consistently achieve 80% accuracy in selecting the correct prompt, even after 10 rounds of incremental learning. We also compare the average accuracy on previously learned categories for each task across different methods as shown in middle figure. We observe a strong correlation between performance and correct prompt selection. For instance, TIPS achieves the highest average accuracy, followed by CODA-Prompt, which aligns with the trend shown in the left figure. Additionally, we report the performance on new class learning, confirming the superior capability of our proposed method in addressing the stability-plasticity dilemma.

(3) **Adaptability analysis.** To further validate our proposal, we combined it with traditional prompt-based CL methods. Specifically, we integrated TP with $\mathcal{L}_{\mathcal{SD}}$ into recent SOTA and tested the Avg and Last-T metrics to assess the adaptability of our proposal within other methods, as shown in Table 6 and Table 7. The results indicate that both modules improve the Avg and Last-T metrics. For instance, it can be observed that there is a notably improvement in AVG following the integration of TP. Subsequently, L2P and CODA-Prompt exhibit respective increases of 5.7% and 6.94% in term of Last-T with the incorporation of $\mathcal{L}_{\mathcal{SD}}$. Overall, it indicate that our method does not require selecting or merging parameters and demonstrates generalizability.

## 5 Conclusion

In this paper, we propose a novel prompt-based continual learning framework without relying on a pre-set database or manually designed prompts. The proposed framework employs a two-level prompt selection strategy, maintaining the stability of prompt selection by learning the consistency between high-level image information and label semantic information in long sequence settings. To achieve a balance between stability and plasticity, we design a semantic knowledge distillation module to reduce the impact of old class retention on the learning of new classes. Extensive experiments on four public datasets demonstrate that our method achieves state-of-the-art performance and is independent of any particular dataset or CL environment.

**Border impact:** The proposed framework learns from a prompt pool and label predictor, making it applicable to downstream tasks such as continual fine-tuning of large language models. However, the goal of our work is to provide a general framework, and the trained prompt pool may be influenced by inherent data biases. Therefore, future work could extend our framework to other application scenarios.

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

# A    SUPPLEMENTARY MATERIALS

## A.1    VISUALISATION OF DETAILED PERFORMANCE

**Visualising Detailed Results.** To compare the performance of our proposed method (TIPS) with the recent SOTA, we provide a detailed report for each dataset (CIFAR, ImageNet-R, CUB, and DomainNet) across three incremental scenarios (5 tasks, 10 tasks, 20 tasks), as shown in Figure 4.

- **CIFAR100:** In the 10 tasks scenario, TIPS has an overall average accuracy that is 2% higher than ESN. However, for both the longer (20 tasks) and shorter (5 tasks) task sequences, our performance is close to SOTA. Compared to attriCLIP, which also uses CLIP assistance, TIPS shows better stability on simpler datasets.

- **ImageNet-R:** This dataset contains multiple domains and classes, presenting greater challenges and requirements for models. Notably, in all three scenarios, our performance exceeds SOTA. We believe this is due to CLIP's powerful cross-domain recognition capabilities.

- **CUB:** This dataset has a distribution that is significantly different from the pre-trained dataset. Therefore, it effectively tests the model's generalization ability for downstream tasks. Although our performance on this dataset is slightly lacking, it is still close to SOTA and shows better resistance to forgetting. For instance, in the 20 tasks scenario, the best-performing Dualprompt experiences a sharp decline in accuracy after incremental session 13, while TIPS maintains stable accuracy and even shows a slight increase.

- **DomainNet:** It is a domain incremental dataset, where TIPS demonstrates strong domain generalization capabilities, outperforming SOTA performance in various scenarios. Notably, this dataset is more challenging than ImageNet-R, leading to a rapid decline in accuracy for all methods after the start of incremental learning. However, TIPS shows less forgetting compared to other methods, proving its robustness in handling this difficulty.

**Hyperparameter sensitivity experiments.** We discuss the length of the parameters for the first-level prompt $m$, the length of the parameters for the second-level prompt $M$, and the setting of the hyperparameter $\lambda$:

In Table 8 and Table 9, we report the impact of different prompt lengths on the final average accuracy. We found that although longer prompts do not lead to higher accuracy, there is a downward trend after a certain point. We believe this effect is mainly because too many parameters make the learned prompts converge, which is not good for retrieval accuracy. Finally, in Table 10, we report the performance of different hyperparameters on different datasets. We select different hyperparameters based on the difficulty of each dataset.

**Visualisation of Grad-CAM. (Selvaraju et al., 2017)** We present the activation images of TIPS and SOTA in Figure 5 (by Grad-CAM). L2p can only focus on a few contours, while Coda-Prompt has a larger focus area than L2p but still shows many unnecessary scattered points. TIPS, on the other hand, can accurately find the target class contours, and the scattered points are significantly reduced.

Table 8: Result of the impact of different first-level prompt $m$ lengths on the 10 tasks incremental accuracy, with the second-level prompt $M$ length fixed at 4.

| First-level Prompt | CIFAR-100 | ImageNet-R | CUB-200 |
|---|---|---|---|
| m=8 | 91.92 | 83.00 | 82.39 |
| m=16 | **92.43** | **83.77** | **84.29** |
| m=32 | 91.93 | 83.66 | 82.93 |

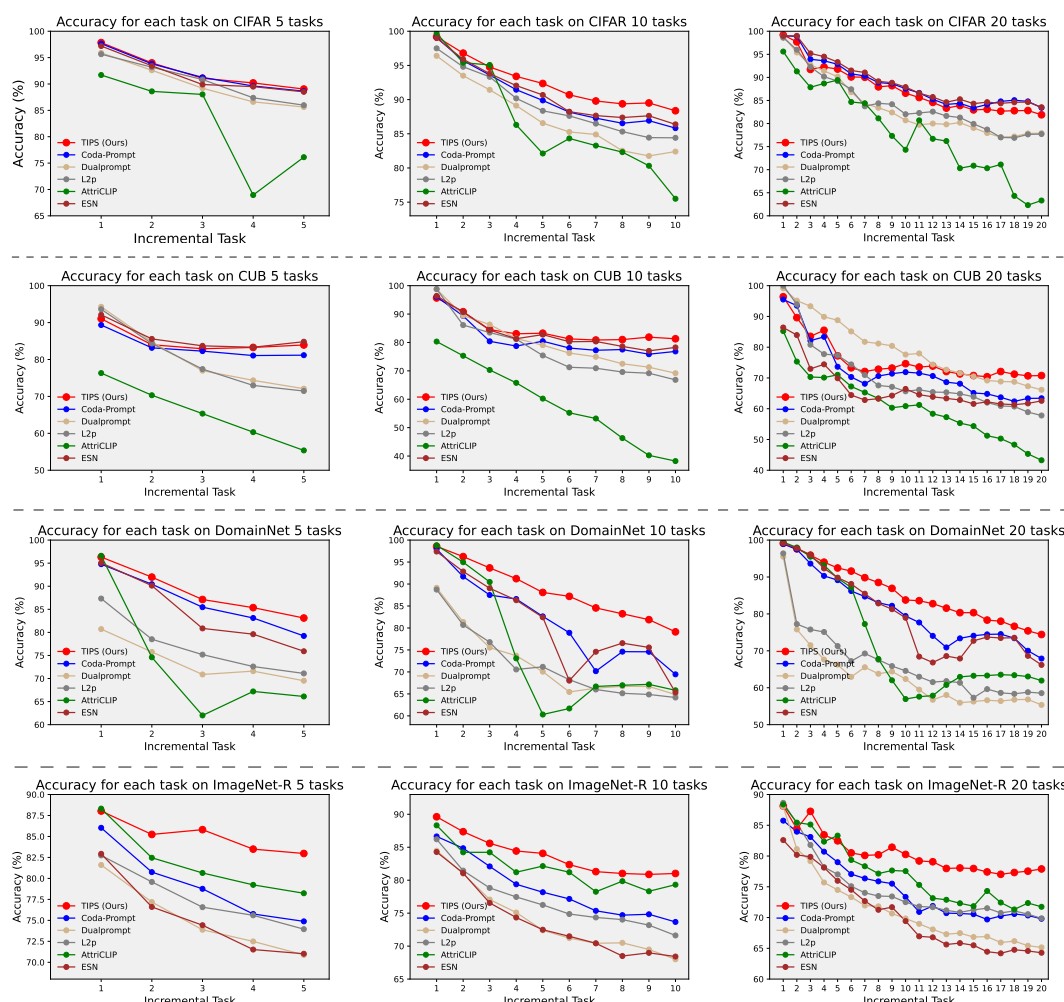

Figure 4: Performance of different methods in continual learning across 4 datasets. Each point represents the accuracy of seen classes.

Table 9: Result of the impact of different second-level prompt $M$ lengths on the 10 tasks incremental accuracy, with the first-level prompt $m$ length fixed at 16.

| Second-level Prompt | CIFAR-100 | ImageNet-R | CUB-200 |
|---|---|---|---|
| M=2 | 92.17 | 82.71 | 82.28 |
| M=4 | **92.43** | **83.77** | **84.29** |
| M=8 | 92.10 | 83.40 | 83.15 |
| M=16 | 91.31 | 83.83 | 83.85 |

Table 10: Result of the Avg with differ $\lambda$ settings on 10 tasks incremental accuracy.

| $\lambda$ | 0 | 0.1 | 0.2 | 0.4 | 1 |
|---|---|---|---|---|---|
| CIFAR-100 | 91.64 | 92.01 | **92.43** | 92.12 | 92.23 |
| ImageNet-R | 83.03 | 83.42 | 83.44 | **83.77** | 83.62 |
| CUB-200 | 82.90 | 83.92 | 84.12 | **84.29** | 84.01 |
| DomainNet | 87.42 | 88.31 | **88.39** | 88.22 | 88.29 |

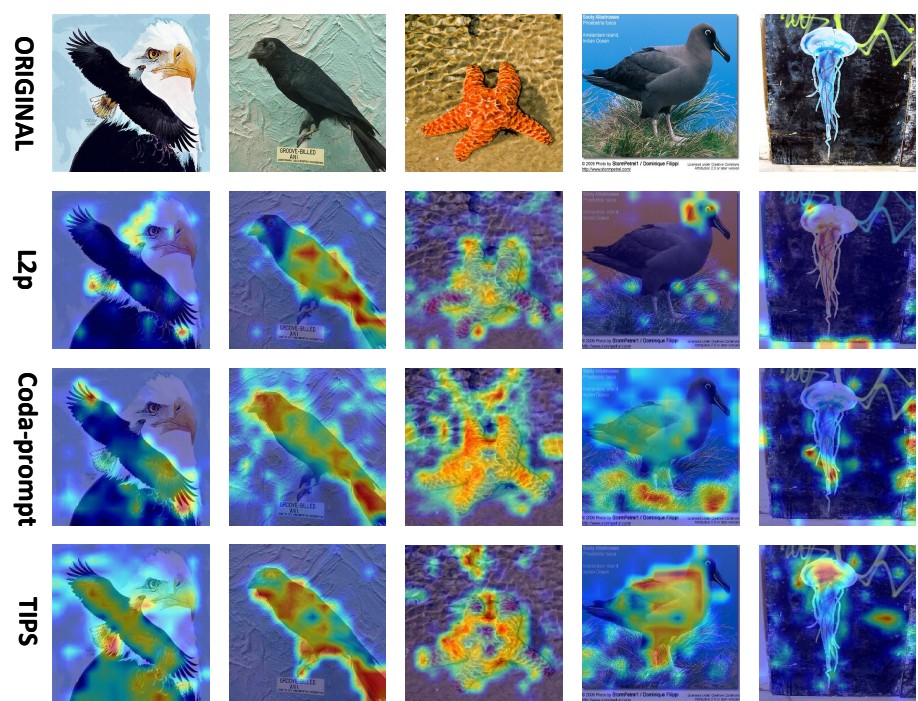

Figure 5: Grad-CAM visualizations of different methods. All experiments were conducted in ImageNet-R 10 tasks setup, with images randomly selected.

---

**Algorithm 1** Two Level Prompt training on $t$-th task.

---

**Input:** Current data stream $(x_i, y_i)_{i=1}^n \in \mathcal{D}_t$, Pre-train Clip-Model $E_{txt}$ and $E_{vis}$, Pre-train
ViT $f_\theta$, classification head $\phi$, First-Prompt Pool $\mathbf{P}^{first} = \left\{ \mathbf{P}_1^{first}, \ldots, \mathbf{P}_t^{first} \right\}$, Second-
level Prompt Pool $\mathbf{P} = \{\mathbf{P}_1, \ldots, \mathbf{P}_t\}$, language token $\mathbf{l} = [\mathbf{l}_1, \mathbf{l}_2, \ldots, \mathbf{l}_D]$, class token
$\mathbf{c} = [\mathbf{c}_1, \mathbf{c}_2, \ldots, \mathbf{c}_D]$, adaptive weight sets $\mathbf{W}$, number of training epochs of $t$-th task $E$, learn-
ing rate $\gamma$, balancing parameter $\lambda$
**Freeze:** $\left\{ \mathbf{P}_1^{first}, \ldots, \mathbf{P}_{t-1}^{first} \right\}$, $\{\mathbf{P}_1, \ldots, \mathbf{P}_{t-1}\}$, $f_\theta, E_{txt}, E_{vis}$

1: Initialize $\mathbf{P}_t^{first}$ with tokenized "XXXX[CLS]"
2: **for** $e = 1, \ldots, E$ **do**
3:      Draw a mini-batch $B = \left\{ (x_i, y_i)_{i=1}^b \right\}$
4:      Initialize the sets of chosen keys and prompts for current batch: $K_B = \{\}, \mathbf{P}_B = \{\}$
5:      **for** $(x, y)$ in $B$ **do**
6:          Calculate query $\mathbf{q} = E_{vis}(B)$ and key $\mathbf{k} = E_{txt}(P_t^{first})$
7:          Retrieve the batch prompt $\hat{\mathbf{P}}_B$ using $\mathbf{k}$ and adaptive weight $\mathbf{W}$ by Eq. 4
8:          Calculate class token embedding $\mathbf{e}^c = f_\theta(x_i)[1]$ and language token embedding $\mathbf{e}^l = f_\theta(x_i)[2]$
9:          Calculate total loss $\mathcal{L}_{total} = \mathcal{L}_{CE} + \lambda \cdot \mathcal{L}_{KD} + \mathcal{L}_O$
10:         Update sets of chosen keys and prompts: $\mathbf{K}_B = \mathbf{K}_B \cup \mathbf{k}, \mathbf{P}_B = \mathbf{P}_B \cup \hat{\mathbf{P}}_B$
11:      **end for**
12:      Accumulate total batches loss $\mathcal{L}_{total}$
13:      Perform backward
14:      Update $\mathbf{P}_t^{first}, \mathbf{P}_t, \mathbf{W}, \mathbf{c}, \mathbf{l}, \phi$ by $\gamma \cdot \nabla \mathcal{L}_{total}$
15: **end for**

---

