# OpenReview forum: "TIPS: Two-Level Prompt for Rehearsal-free Continual Learning"
_ICLR.cc/2025/Conference — ICLR 2025 Conference Withdrawn Submission_

### Official Review · Reviewer_Hp5t · 2024-10-27

**Soundness:** 2
**Presentation:** 1
**Contribution:** 2
**Rating:** 3
**Confidence:** 4

**Summary:**

The paper introduces TIPS, a prompt-based continual learning (CL) framework that incorporates a two-level prompt selection strategy with adaptive weights to enhance both accuracy and stability across long task sequences. Additionally, a semantic knowledge distillation module improves generalization to new classes by leveraging high-level image and label semantics. Experimental results on four benchmark datasets (CIFAR-100, ImageNet-R, CUB-200, and DomainNet) show that TIPS outperforms state-of-the-art methods.

**Strengths:**

1. The proposed two-level prompt selection strategy effectively improves prompt selection accuracy according to the ablation study.

2. TIPS outperforms the baseline methods across four benchmark datasets, demonstrating its effectiveness.

**Weaknesses:**

1. Some statements are unclear. For example, in the sentence “It requires training only a few parameters to quickly adapt to downstream tasks”, it is unclear what “it” refers to. Similarly, the sentence “Compared to other traditional continual learning methods, it is more effective in resisting catastrophic forgetting” lacks precision in referring to which aspect or component is responsible for this benefit. Please clarify which specific method or component they are referring to in these sentences. This would help improve the overall clarity of the paper.

2. In the introduction, the discussion on previous CL methods that train models from scratch is not entirely accurate. The authors are encouraged to refer to comprehensive survey papers such as “A Continual Learning Survey: Defying Forgetting in Classification Tasks” (TPAMI, 2021) and “A Comprehensive Survey of Continual Learning: Theory, Method, and Application” (TPAMI, 2024) to strengthen their understanding and presentation of related CL methods.

3. While the paper uses density distribution changes (visualized in Fig. 3 (b)) to demonstrate the stability of the proposed method, additional quantitative measurements for density distribution shift would provide stronger support for the claim. Please consider using KL divergence or other quantitative metrics to measure density distribution shifts.

4. In related work, the comparison with prompt-based methods is not entirely clear. For example, the statement that deciding “which parameters to merge” is a drawback of CLIP-based methods is vague. It is unclear what “merging parameters” entails in these contexts. Furthermore, the paper does not discuss the drawbacks of methods that use Transformers as backbones, leaving a gap in the comparison.

5. The authors wrote that, compared to CLIP-based approaches, TIPS does not involve “fine-tuning the backbone network” and “does not require additional storage space for old class instances due to the use of a pre-trained model's prior knowledge”. However, these claims are inaccurate, as CLIP-based methods also do not suffer from these limitations. Please clarify how TIPS differs from CLIP-based methods in these aspects, if at all.

6. The paper would benefit from thorough revision to enhance the writing. Below are some examples of sentences that could be improved: “CL aims at smoothly integrating new tasks into a single model while preventing catastrophic forgetting of previously learned knowledge” and “In recent, some works …”, among others.

**Questions:**

The authors state that existing CL methods “focus on solving the model’s stability issues”. Why, then, do these methods still exhibit larger distribution changes compared to the proposed approach, as shown in Fig. 3 (b)?

---

### Official Review · Reviewer_zbyv · 2024-11-03

**Soundness:** 3
**Presentation:** 4
**Contribution:** 3
**Rating:** 5
**Confidence:** 5

**Summary:**

Prompt selection is a critical component in the framework of prompt-based continual learning. This work designs a two-level prompt selection strategy that leverages the semantic consistency between images and class labels to improve the corresponding prompt's accuracy. Moreover, a semantic distillation module is proposed to integrate the visual and linguistic modalities to utilize the semantics of the original class labels in text form. Evaluation of several benchmarks verifies the effectiveness of the proposed method.

**Strengths:**

-The proposed method is clear and easy to understand

-Extensive evaluation verifies the effectiveness of the proposed method.

**Weaknesses:**

1、The motivation is not clear. For example, why can using the two-level prompt obtain a stable prompt selection strategy and improve the plasticity?

2、Limited novelty. The proposed method consists of two-level prompts. The first-level prompt is the same as the context optimization introduced in CoOp, and the second-level prompt is an additional prompt with similar scores. Therefore, the proposed two-level prompt has limited contribution.

3、The proposed method is most similar to the Semantic Two-level Additive Residual Prompt[R1] published in Arxiv on Mar 2024 and accepted by ECCV 24.  It is recommended that the authors explain the differences between the two papers.
[R1] Semantic Residual Prompts for Continual Learning, ECCV 24

4、L262: What is the definition of the language and class tokens?

5、The semantic knowledge distillation \mathcal{L}_{KD} aims to constrain the consistency between the class-level embedding h_c generated by the Text Encoder and the e^{l}_{c} generated by the Visual Encoder, whose goal seems to constrain the generate visual class proto is consistent with the textual class proto. However, why not use the standard constrastive loss introduced in CLIP? The motivation seems to be not reasonable.

6、As shown in Table 1, the experiments are conducted with three random seeds: 1993, 1997, and 1999. Why use those three specific seeds?

**Questions:**

Please see #Weaknesses

---

### Official Review · Reviewer_vmHv · 2024-11-04

**Soundness:** 3
**Presentation:** 2
**Contribution:** 2
**Rating:** 5
**Confidence:** 4

**Summary:**

The paper introduces TIPS, a prompt-based CL method designed to improve the stability and plasticity of prompt selection for rehearsal-free continual learning. TIPS includes a two-level prompt selection strategy and a semantic distillation module, enhancing both prompt selection accuracy and model adaptability across diverse tasks. Experimental results show that TIPS outperforms other methods across multiple datasets.

**Strengths:**

The primary strength of this work lies in its strong performance. TIPS consistently outperforms existing methods in both accuracy and forgetting rate across multiple datasets, demonstrating high adaptability and resilience against catastrophic forgetting. Another strength is its integration of information from both textual and visual tokens, enhancing similarity alignment and improving prompt selection accuracy.

**Weaknesses:**

There are several weakness:
1) The integration of a multi-level prompt selection and adaptive weighting strategy could increase computational requirements, potentially limiting practical application in resource-constrained environments.
2) The use of CLIP’s text and image encoders, which contain extensive pre-trained knowledge, raises concerns about data overlap. The testing data may overlap with or be highly correlated to CLIP’s training data, making the observed performance gains somewhat expected. It’s unclear if the improvement is due to the novel aspects of TIPS or simply the inclusion of CLIP, as other methods might similarly benefit from using CLIP. This ambiguity makes it unclear to identify the key design elements driving performance improvement.
3) It is confusing that, in the experiments L340 it says using the Pre-Trained model on ImageNet-21K but on the Figure2 are all using CLIP models. Also, the performance results reported in Table 1 differ significantly from those in the original papers, such as CODA-Prompt, making it difficult to interpret the findings accurately.

**Questions:**

A few questions could be clarified for a better interpretation of this paper:
1) How does TIPS perform in real-world settings with varied task distributions? Specifically, how well does it handle testing datasets that largely differs from CLIP’s training data, such as medical images or fine-grained datasets like iNaturalist? This would help to understand whether the performance gains are primarily due to the CLIP model and assess TIPS’s robustness across diverse tasks.
2) What is the computational overhead introduced by TIPS compared to other methods? Additionally, how does TIPS perform when compared under an equivalent computational budget—one setup with TIPS and another with a scaled-up model size, such as from ViT-B to ViT-L?
3) Can TIPS be applied to open-world scenarios [1] [2]? For instance, how would it handle a test example with a class name that was never present in the training dataset?

[1] Open-world continual learning: Unifying novelty detection and continual learning, AIJ 2024
[2] Open-World Dynamic Prompt and Continual Visual Representation Learning, ECCV 2024

---

### Official Review · Reviewer_a4u6 · 2024-11-09

**Soundness:** 3
**Presentation:** 3
**Contribution:** 2
**Rating:** 5
**Confidence:** 5

**Summary:**

The topic of this paper is about rehearsal-free Continual Learning (CL) task with prompt-based methods. The authors focus on the prompt selection stage and appropriate regularization terms on prompts learning. They propose a TIPS CL method, which consists of a two-level prompt selection strategy and a semantic knowledge distillation module. The proposed method has been evaluated on several benchmarks.

**Strengths:**

+ The paper writing is good and easy to follow.
+ The proposed method achieves performance gain on several benchmarks.
+ Exploring the prompt-selection and prompt regularization methods on CIL task is interesting.

**Weaknesses:**

- Some works (e.g. [a][b]) have discussed the stability and plasticity trade-off in the prompt-based CL scenario. As described in this paper, the proposed method also considers both the forgetting problem (stability) and generalization ability (plasticity). To better show the effectiveness of this work, making a performance comparison to these methods is necessary.

[a] LW2G: Learning Whether to Grow for Prompt-based Continual Learning, arxiv2024

[b] Hierarchical Decomposition of Prompt-Based Continual Learning: Rethinking Obscured Sub-optimality, NeurIPS 2023

- Some of prompt-based CIL methods dynamically generate prompts, and these important related works are not discussed and compared in this paper. As described in this paper, the two-stage “select+prediction” prompt-based CIL methods usually meet the challenge of prompt select stage. However, some “prompt-generation+prediction” prompt-based CIL methods (e.g. [c]-[d]) directly generate instance-wise prompts and do not need to select prompts at the first stage. It is necessary to give a detailed analysis about these methods. For instance, conduct experiments on CIL benchmarks with “prompts generation” or “prompts selection” pipelines. In different CIL scenarios, we need to know how to choose these two pipelines. Can the proposed two-level prompt selection strategy achieve better performance than prompt generation strategy?

[c] Generating Instance-level Prompts for Rehearsal-free Continual Learning, ICCV2023

[d] Open-World Dynamic Prompt and Continual Visual Representation Learning, ECCV2024

- Comparing to prompt-regularization CL methods (e.g. [e]) on prompt learning. The proposed semantic knowledge distillation module is utilized to regularize the learning of prompts on new tasks. To better show the effectiveness of this module, it is necessary to compare the performance of these methods and highlight the difference between these methods.

[e] OVOR: One Prompt with Virtual Outlier Regularization for Rehearsal-Free Class-Incremental Learning, ICLR2024

**Questions:**

My major concerns come from the important experimental comparison to related works and the discussion about these works. These analysis and experiments comparison among these works are necessary to illustrate the contribution of this work. The details are included in the above weaknesses.

---

### Note · Authors · 2025-01-12

**Comment:**

I would like to take this opportunity to sincerely thank the reviewers and the program committee for their valuable feedback and effort during the review process. Their insights have been instrumental in helping me identify areas for improvement, and I greatly appreciate the time and expertise they dedicated to reviewing my submission.

**Withdrawal Confirmation:**

I have read and agree with the venue's withdrawal policy on behalf of myself and my co-authors.